# Targeting TRAIL Death Receptors in Triple-Negative Breast Cancers: Challenges and Strategies for Cancer Therapy

**DOI:** 10.3390/cells11233717

**Published:** 2022-11-22

**Authors:** Manjari Kundu, Yoshimi Endo Greer, Jennifer L. Dine, Stanley Lipkowitz

**Affiliations:** Women’s Malignancies Branch, Center for Cancer Research, National Cancer Institute, National Institutes of Health, Bethesda, MD 20892, USA

**Keywords:** TRAIL, death receptors, apoptosis, triple-negative breast cancer, immunomodulation

## Abstract

The tumor necrosis factor (TNF) superfamily member TNF-related apoptosis-inducing ligand (TRAIL) induces apoptosis in cancer cells via death receptor (DR) activation with little toxicity to normal cells or tissues. The selectivity for activating apoptosis in cancer cells confers an ideal therapeutic characteristic to TRAIL, which has led to the development and clinical testing of many DR agonists. However, TRAIL/DR targeting therapies have been widely ineffective in clinical trials of various malignancies for reasons that remain poorly understood. Triple negative breast cancer (TNBC) has the worst prognosis among breast cancers. Targeting the TRAIL DR pathway has shown notable efficacy in a subset of TNBC in preclinical models but again has not shown appreciable activity in clinical trials. In this review, we will discuss the signaling components and mechanisms governing TRAIL pathway activation and clinical trial findings discussed with a focus on TNBC. Challenges and potential solutions for using DR agonists in the clinic are also discussed, including consideration of the pharmacokinetic and pharmacodynamic properties of DR agonists, patient selection by predictive biomarkers, and potential combination therapies. Moreover, recent findings on the impact of TRAIL treatment on the immune response, as well as novel strategies to address those challenges, are discussed.

## 1. Introduction

Triple-negative breast cancer (TNBC) is defined by the absence of estrogen receptor, progesterone receptor, and human epidermal growth factor receptor 2 (HER2) amplification and accounts for ~15% of breast cancer cases [1]. It has the worst prognosis of the major subtypes of breast cancer due in part to the absence of well-defined molecular targets [1]. TNBC responds to chemotherapy, but metastatic relapse of the treated early stage is frequent, and there are no curative therapies in the advanced stage [1]. While new targeted therapies (e.g., trophoblast antigen 2 [trop2]-targeted antibody-drug conjugates [ADCs]) and immune checkpoint inhibitors have significant activity for the treatment of TNBC [2,3,4,5,6], novel therapies are still needed for TNBC especially in the advanced setting. In this review, we will describe the tumor necrosis factor-related apoptosis-inducing ligand (TRAIL also known as Apo2L/TNFSF10) pathway, and the data supporting a role for TRAIL receptor agonists in the treatment of TNBC.

TRAIL is a member of the tumor necrosis factor (TNF) family of ligands capable of initiating apoptosis through engagement of its death receptors. TRAIL binds to Death Receptor 4 (DR4, also known as TRAIL-R1/TNFRSF10A) and Death Receptor 5 (DR5, also known as, TRAIL-R2/TRICK-2/KILLER/TNFRSF10B), causing caspase-mediated apoptosis (Figure 1A) [7,8]. TRAIL is mainly expressed on the cell surface of immune cells eliciting programmed death of target cells and has been reported to induce apoptosis in a variety of cancer cell lines while sparing the normal cells [7,9,10,11,12]. However, there are some reports of TRAIL-induced apoptosis in normal cells, such as primary salivary epithelial cells [13], prostate epithelial cells [14,15] and primary human epithelial esophageal cells [16]. Preclinical animal studies supported the lack of overt toxicity of TRAIL to the normal tissues [7,17,18]. Therefore, targeting TRAIL/DR pathway offers an attractive approach to induce apoptosis in cancer cells. Early attempts to treat cancer using the TNF and Fas ligand (FasL) as DR agonists showed disappointing results due to lack of efficacy or prohibitive preclinical toxicity [7,17,18]. This has prompted investigation into the use of TRAIL or DR4/5 agonist antibodies in cancer therapy. TRAIL agonist induced regression of cancer xenografts in mice without affecting normal tissues, and human phase 1 studies have demonstrated that TRAIL agonists are safe and well tolerated in patients [7,17,18,19,20,21,22,23]. However, phase 2/3 clinical trials utilizing recombinant human (rh)TRAIL and agonistic antibodies directed at DR4/5, while well-tolerated, have not shown significant clinical efficacy [23,24,25].

## 2. TRAIL and Its Receptors

TRAIL is a type II transmembrane protein that undergoes proteolytic cleavage to produce an extracellular ligand [26]. TRAIL shares structural homology with other TNF superfamily members, including TNF and FasL (~25% homology), and can potently induce apoptosis in its soluble and membrane-bound forms [12,27,28]. Similar to other members of the TNF family, active TRAIL is comprised of a homotrimer formed by three TRAIL monomers and is further characterized by a zinc-binding site located near the trimerization interface that is necessary for maintaining the stability and activity of the protein [29]. Zinc depletion by chelation induces TRAIL dimerization that results in dramatically decreased ability to bind and activate DRs [29]. In human cells, TRAIL induces apoptosis by activating DR4 and DR5, members of the TNF receptor family [30]. In mice, only one functional apoptosis-inducing TRAIL receptor is expressed, called murine TRAIL Receptor (mTRAIL-R) [31]. DR4 and DR5 possess a death domain (DD) that allows for the transduction of the apoptotic signal through the recruitment of adaptor proteins and caspase-8 to facilitate formation of the death-inducing signaling complex (DISC) (Figure 1A) [7,32]. TRAIL also can bind to decoy receptor 1 (DcR1) which lacks a DD altogether, and DcR2 which has a truncated DD. These decoy receptors are unable to induce DISC formation and act as negative regulators of the apoptotic signaling by competitively binding TRAIL [7,10,33]. Osteoprotegerin is a soluble protein that also binds to TRAIL and inhibits activation of DR4 and DR5 in experimental models [34]. Osteoprotegerin competes with Receptor Activator of Nuclear Factor Kappa-B Ligand (RANKL) for binding to the receptor RANK and thereby inhibits osteoclast differentiation and bone resorption [32]. Findings from studies have been mixed regarding whether TRAIL interferes with the anti-osteoclast activities of osteoprotegerin [35,36,37,38,39]. However, the loss of TRAIL was not found to affect bone density in vivo, suggesting that the interaction between TRAIL and osteoprotegerin may not impact RANK signaling with respect to bone development or turn-over [36]. Whether osteoprotegerin affects TRAIL signaling under physiological conditions is still unclear [40].

The TNF super family members have been well described as important modulators of immune cell function [41]. TRAIL mRNA is expressed in many tissues types [12,27] as well as on B cells, T cells and monocytes play important roles in a wide variety of innate and adaptive immune responses [41,42]. Stimulated natural killer (NK) cells, T cells, and dendritic cells demonstrate upregulation of TRAIL [41,42]. Importantly, TRAIL is upregulated in response to interferon (IFN)γ in NK cells and serves as an effector of NK cell function in vitro and in vivo [42]. TRAIL upregulation in response to IFNγ suppressed primary experimental tumors and metastasis and has also been found on IFNγ-producing dendritic cells [42]. Ablation of TRAIL results in thymocyte apoptosis and increased autoimmune responses in mice [43,44]. Moreover, TRAIL depletion resulted in enhanced experimental and spontaneous metastasis in mice [45]. These findings demonstrate that TRAIL provides critical immune surveillance and regulatory functions, including the suppression of autoimmunity and inhibiting tumor growth and metastasis.

## 3. TRAIL Signaling Pathways

Activation of TRAIL death receptors by their cognate ligands induces apoptosis but as will be described below, under certain circumstances, activation of the TRAIL death receptors can promote growth and in tumors induce metastasis or pro-tumorigenic immune effects. The canonical TRAIL-induced apoptotic signaling pathway is an example of apoptosis mediated through the extrinsic death pathway, which entails activation of cell-surface receptors by a ligand to induce activation of downstream caspases (Figure 1A) [30]. Activation of DR4 and DR5 by TRAIL promotes receptor clustering and formation of the DISC [46]. The adaptor protein, Fas-associated death domain protein (FADD), also contains a death domain (DD) which interacts with the DD of DR4/5 in a homotypic fashion [46]. FADD also contains a death effector domain (DED) which recruits pro-caspases-8 and -10 via a homotypic interaction with DED of the pro-caspases [46]. The forced proximity of the pro-caspases-8 and -10 at the DISC leads to auto-processing of the pro-caspases [46], resulting in an active tetramer of two large and small subunits [46]. This results in activation of downstream caspases such as caspase-3 or -7, ultimately inducing apoptosis. FLIP (FLICE [FADD-like IL-1β-converting enzyme]-inhibitory protein), a negative regulator of the TRAIL pathway, is structurally related to pro-caspases-8 and -10 with N-terminal DED domains, and a C-terminal caspase-like domain, but the catalytic cysteine is replaced by tyrosine, thus rendering it catalytically inactive [47]. FLIP may also be recruited to the DISC, prevent caspase-8 or caspase-10 from interacting with FADD, and thus attenuate the apoptotic signal [46,47].

In some cells, activated caspase-8 or -10 can cleave Bid, a pro-apoptotic BH3-only BCL-2 family protein, into truncated Bid (tBid), which translocate to the mitochondria and activates the intrinsic death pathway (Figure 1A) [48,49,50]. The BCL-2 protein family regulates the intrinsic death pathway via anti- and pro-apoptotic family members [49]. When the intrinsic death pathway is activated, pro-apoptotic BCL-2 family members, such as BAD, BIM, and PUMA, antagonize anti-apoptotic family members, including BCL-2, BCL-xL, BCL-w, and MCL1 [49,51]. tBid directly and indirectly activates pro-apoptotic proteins Bak and Bax, causing mitochondrial outer membrane permeabilization (MOMP) and cytochrome c release [34,48,49,52]. The scaffold protein apoptotic protease-activating factor 1 (APAF1) binds to cytochrome c and activates caspase-9 [48,49]. Second mitochondrial activator of caspases (SMAC), an inhibitor of apoptosis proteins (IAPs) that suppress caspase function, is also released during MOMP to help facilitate apoptosis [53]. The extrinsic and intrinsic death pathways converge; caspases-8, -10, and -9 mediate proteolytic processing of the executioner caspases-3 and -7 which carry out the final steps of apoptosis by cleaving numerous substrates [48,49]. Activated caspase-3 is also able to activate caspase-8 in a feedback-loop, thus amplifying the apoptotic signal [54,55].

Like TNF, TRAIL-induced DR4/5 activation has also been associated with the induction of nuclear factor-κB (NF-κB) and mitogen-activated protein kinase (MAPK) signaling [47,56,57,58] (Figure 1B). TNF signaling is a more potent activator of these pathways than TRAIL [57]. After TRAIL-induced caspase activation, receptor-interacting serine/threonine-protein kinase 1 (RIPK1/RIP1), FADD, TNF receptor-associated factor 2 (TRAF2), and caspase-8 form a secondary complex that facilitates kinase signaling in a RIPK1-dependent manner [57]. However, findings have been mixed concerning the anti- and pro-survival mechanisms induced by TRAIL-mediated NF-κB and MAPK activation [40,59]. Inhibition of components of the NF-κB and MAPK signaling pathways has produced examples of both enhanced and attenuated sensitivity to TRAIL-induced apoptosis, demonstrating that the outcomes associated with TRAIL-mediated activation of NF-κB and MAPK will vary depending on context and requires further characterization [40,59].

TRAIL signaling is also associated with non-apoptotic cell death mechanisms, including the caspase-independent, cell-regulated form of necrosis, necroptosis [59,60,61,62,63] (Figure 1C). TRAIL has been found to induce necroptosis regulators RIPK1 and RIPK3 in an acidic pH-dependent manner, and NF-κB inhibition has been found to enhance sensitivity to TRAIL-induced necroptosis [59,60,62,63]. Mechanistically, necroptosis depends on activation of RIPK1 and RIPK3 and necrosome complex formation involving RIPK1 and RIPK3 and the mixed lineage kinase domain-like protein (MLKL) [64] (Figure 1C). Recently, a study indicated the involvement of an E3 ubiquitin-protein ligase TRIM21(tripartite motif containing 21) in endogenous TRAIL-mediated necrosome formation [65]. Further elucidation of the mechanisms and conditions under which TRAIL activates necroptotic signaling is needed.

Importantly, numerous studies show that TRAIL can also induce autophagy [66]. Autophagy and apoptosis are both important cellular processes controlled by distinct groups of regulatory mechanisms [67]. They also have a crosstalk to regulate each other [68,69,70]. As shown in Figure 1D, autophagy and apoptosis share same regulatory factors, including the Bcl-2 family [49,69,71] and FLIP [72]. Bcl-2 family members inhibit Beclin-1 [73,74] which is required for activation of autophagy [75]. FLIP limits the ATG3-mediated LC3 conjugation to inhibit autophagosome biogenesis [72]. Caspase-8 has been shown to regulate autophagy by targeting autophagic components, such as ATG3, ATG5 and Beclin-1 [76,77,78,79]. Autophagosome formation can facilitate caspase-8 activation by providing a platform consisting of ATG5 and LC3 in some contexts [80], while sequestration of pro-caspase-8 to the autophagosome can lead to either caspase-8 induction, or downregulation [81]. The importance of autophagy in TRAIL resistance breast cancer cells is still being studied intensively. Notably, in different models of breast cancer cell lines resistance to TRAIL-induced killing was attributed by TRAIL induced autophagy [82]. Furthermore, in a series of breast cancer cell lines autophagy was shown to have an impact on dynamics of TRAIL receptors. In these cell lines, autophagosome accumulation led to decrease in TRAIL-induced death by downregulating the TRAIL receptors [83]. While these data suggest that TRAIL-induced autophagy inhibits TRAIL-mediated death, there are descriptions of TRAIL-induced autophagic cell death. For example, in a 3D culture model of breast lumen formation using the immortalized but not transformed human mammary MCF-10A cell line, TRAIL-induced autophagy contributed to the death of the luminal cells that led to hollow lumen formation [84]. Thus, understanding TRAIL-induced autophagy is required to decipher TRAIL mediated response or resistance.

TRAIL and DR expression have been shown to positively regulate cell growth under certain conditions. Specifically, TRAIL signaling promotes proliferation and IFNγ production in pre-activated T cells [85]. TRAIL-induced activation of DR5 and mTRAIL-R signaling is correlated with enhanced TNBC-associated bone metastasis and KRAS-driven lung and pancreatic cancer metastasis, respectively [86,87]. The first study utilized a bone tropic variant of the TNBC MDA-MB-231 cell line which is known to contain an oncogenic KRAS mutation which is infrequent in breast cancer [88]. In this study the investigators found that DR5 was upregulated in the bone tropic cells compared to the parental MDA-MB-231 cell line and they demonstrate that RNAi mediated knockdown of DR5 reduces the levels of bone metastasis related genes (e.g., HMGA2, phosphor-Src, and the C-X-C cytokine receptor 4) in the cancer cell. When injected intracardially, the knockdown of DR5 reduced the ability of the cells to metastasize to bone [86]. Thus, the pro-metastatic phenotypes were reversed with DR5 and mTRAIL-R inhibition, suggesting that inhibition of the TRAIL signaling may provide a novel method to inhibit metastasis.

As discussed above, TRAIL knockout mice have shown increased metastases [45]. Additional characterization of the conditions under which TRAIL signaling is inhibitory or pro-proliferative or pro-metastatic is necessary to determine the contexts wherein TRAIL pathway activation or inhibition are the appropriate therapeutic strategy in TNBC.

## 4. Preclinical Data of TRAIL Death Receptor Agonists in Breast Cancer

Early studies evaluating TRAIL agonist-induced apoptosis in breast cancer cell lines revealed that while the MDA-MB-231 cell line was sensitive to the rapid induction of caspase dependent apoptosis, most of the cell lines tested were resistant to TRAIL-mediated apoptosis [26,58,89,90,91]. Subsequent work evaluating a wider panel of breast cancer cell lines with different phenotype (e.g., hormone receptor positive, HER-2 amplified, or triple-negative cell lines) demonstrated that TRAIL agonists preferentially induced apoptosis in TNBC (including MDA-MB-231) [30]. In this study, eight of eleven triple-negative breast cancer cell lines were highly sensitive to TRAIL-induced apoptosis while cell lines representing hormone receptor positive or HER2 amplified breast cancer were either resistant or only moderately sensitive. These findings were confirmed in additional studies [89,92,93,94]. Further evaluation of the TRAIL agonist-sensitive TNBC revealed that the most sensitive TNBC cell lines have a mesenchymal phenotype which are designated as basal B [30,94,95,96]. This will be discussed further in Section 6 below.

Most TNBC cell lines express both the agonistic TRAIL receptors DR4 and DR5 but based on RNAi targeting DR4 or DR5 and on receptor selective mutants of TRAIL, DR5 is the predominant DR required for TRAIL agonist-induced apoptosis in TNBC [30,93,94,97]. One observation that may explain the predominance of DR5 compared to DR4 is that DR5 has a significantly higher affinity (Kd ≤ 2 nM) for TRAIL than DR4 (Kd = 70 nM) [98]. Another possibility is that while both DR4 and DR5 are expressed on most of the TNBC cells tested, the absolute protein levels of DR5 may be higher than DR4, although this has not yet been directly tested.

An important observation is that TRAIL can induce apoptosis in cells that are mutant for p53 [99] and TNBCs frequently have mutation or loss of p53 [100,101]. Loss of p53 function is associated with resistance to chemotherapy and radiation therapy in many cancers, including breast cancer [102,103,104,105,106]. Intriguingly, most of the TRAIL-sensitive triple-negative breast cancer cell lines have mutated or lost p53 [30,95] which implies that TRAIL death receptor agonists may overcome the intrinsic chemotherapy resistance in TNBC deficient in p53.

These preclinical observations of TRAIL sensitivity in breast cancer led to early phase studies of TRAIL death receptor agonists in breast cancer. The following sections of this review will focus on the various clinical trials with TRAIL death receptor agonists, and then focus the remainder of the review on ways to improve the efficacy of these agents.

## 5. Clinical Results with TRAIL Death Receptor Agonists

### 5.1. Clinical Studies

Based on promising preclinical data in breast and multiple other cancer types, DR4 and DR5 agonists have been developed for clinical use over the past two decades, and, as outlined below, trials of single agent TRAIL death receptor agonists and combinations have been largely disappointing. In this section we will summarize the clinical studies across all tumor types and the one study that was TNBC specific. The majority of TRAIL DR agonists are antibodies, including the DR4-specific agonist mapatumumab [107] and the DR5-specific agonists conatumumab [108], drozitumab [109], lexatumumab [110], LBY135 [111], and tigatuzumab [112]. Dulanermin, a form of rhTRAIL that activates both DR4 and DR5 [7], and TAS266, a DR5 agonist tetravalent Nanobody^®^ [113], have also been developed for use in clinical trials [114].

The clinical trials targeting TRAIL DR pathway are listed in Table 1.

Phase 1 dose-escalating monotherapy studies have been carried out in children and adults with advanced solid tumors and adults with non-Hodgkin’s lymphoma (NHL) [19,20,21,113,115,116,117,118,119,120]. All the TRAIL DR agonist antibodies as well as dulanermin were well-tolerated. Dose-limiting toxicities varied, and the maximum tolerated dose was never reached in most studies using these TRAIL DR agonists. Only TAS266 showed toxicity resulting in early termination of the phase 1 trial [113]. Hepatotoxicity (grade ≥ 3 elevations in aspartate aminotransferase and/or alanine aminotransferase levels) and immunogenicity to TAS266 were observed in three of four patients [113]. The presence of pre-existing antibodies against TAS266 in the patients who experienced hepatotoxicity, possible elevated expression of hepatic DR5, and high potency of the TAS266 have been proposed as the potential basis for the hepatotoxicity observed in these patients [113].

Despite the good tolerance, TRAIL DR agonists have demonstrated no or only modest therapeutic benefit in the clinical trials. Most patients treated with a TRAIL DR agonist progressed on treatment or had stable disease (SD). Very few patients experienced sustained partial responses (PR). Two patients with chondrosarcoma achieved a PR while treated with dulanermin and continued to receive treatment ~3 and ~4 years from the initiation of treatment [19]. In another study, one patient with non-small cell lung cancer (NSCLC) maintained a PR ~4 years after the initiation of conatumumab treatment [121]. Although biomarkers to assess for drug pharmacodynamic activity in tumor samples (e.g., activated caspase-3) were evaluated [121] they did not appear to be correlated with tumor response. Currently, a controlled Phase 3 study with Circularly Permuted TRAIL (CPT), another type of recombinant TRAIL variant, is recruiting patient with multiple myeloma (ChiCTR-IPR-15006024).

The combination of TRAIL DR agonists with other agents has been explored in phase 1b and 2 studies (Table 1). Dulanermin and conatumumab were each separately combined with FOLFOX6 and bevacizumab for the treatment of metastatic colorectal cancer [122,123], and dulanermin was combined with paclitaxel, carboplatin, and bevacizumab for the treatment of NSCLC (NCT00315757) [22]. Conatumumab was also combined with another investigational drug, ganitumab (a human monoclonal antibody against type 1 insulin-like growth factor receptor), for the treatment of advanced solid tumors [124]. The addition of TRAIL DR agonists to the treatment regimens was well-tolerated, and drug–drug interactions were not reported. However, the addition of a TRAIL DR agonist did not significantly improve outcomes.

DR agonists have also been studied in phase 2 studies (Table 1). A single complete response (CR) was observed in the first-line treatment of NSCLC with the combination of the DR5 agonist conatumumab, paclitaxel, and carboplatin [125]. Interestingly, this response occurred in the arm of the study in which the lowest dose of conatumumab was tested (3 mg/kg vs. 15 mg/kg every 3 weeks). Another CR was achieved using the combination of the DR4 agonist mapatumumab, paclitaxel, and carboplatin for NSCLC [126], and a CR was observed in a single-agent trial using mapatumumab for the treatment of NHL [23]. Ongoing, phase 1/2 immunotherapy-related clinical trials involving CAR-T cells and DR5 agonist are also reported (NCT03941626).

There has only been one phase II clinical trial reported in patients with breast cancer to date. This trial randomized patients with metastatic TNBC to either albumin bound paclitaxel (nab-PAC) alone or in combination with the DR5 agonist antibody tigatuzumab [127]. A total of 3 CRs out of 39 patients were observed in TNBC patients who received tigatuzumab and nab-PAC [127]. No CRs were observed in any of the 21 patients who received nab-PAC alone, but the overall response rate and Progression-Free Survival (PFS) were similar in both treatment and control arms. Despite rare instances where a CR was achieved, the findings from this phase 2 studies indicate that the addition of a TRAIL DR agonist did not add any significant clinical benefit to unselected TNBC patients or in other cancers [22,23,122,124,125,126,127,128,129,130,131,132,133].

### 5.2. Evaluating Pharmacokinetic and Pharmacodynamic Characteristics of TRAIL DR Agonists in Patients Clinical Studies

In vitro and in vivo preclinical studies have demonstrated the efficacy of TRAIL DR agonists alone or in combination with other agents to selectively induce apoptosis in cancer cells [109]. However, results from clinical trials have largely failed to show significant activity of TRAIL DR agonists. Evaluating the pharmacokinetic and pharmacodynamic properties of TRAIL DR agonists may provide some insight into the lack of clinical activity.

Dulanermin, a form of the human TRAIL protein [7], has a short half-life of ~30 min to 1 h and was administered at 0.5 to 30 mg/kg/day for 3 to 5 consecutive days every 2 to 3 weeks [19,22,123]. Its short half-life may partially explain the observed lack of effectiveness when using dulanermin in advanced tumors. The TRAIL DR antibody agonists have demonstrated a considerably longer half-life of approximately 10 days to 2 weeks [19,20,21,115,117,118,119]. However, DR antibody agonists again had little activity; prolonged half-life may not address all the issues that led to poor clinical activity.

Cross-linking of drozitumab (a TRAIL DR5 selective agonist antibody) by both activating and inhibitory Fcγ receptors (FcγR) on leukocytes was found to enhance apoptosis in vitro and in vivo, although in other targeted antibody agents, such as rituximab and trastuzumab, engagement specifically with activating FcγR was necessary for activity [134]. Similarly, conatumumab requires crosslinking to facilitate apoptosis [108]. Thus, the presence or absence of infiltrating immune cells that express FcγR may impact the effectiveness of DR antibody agonists in humans. Moreover, variants in FcγR, specifically FcγRIIA131H and FcγRIIIA158V, have been associated with enhanced therapeutic efficacy of rituximab, trastuzumab, and cetuximab in humans [134,135,136,137,138,139,140]. Of note, FcγR polymorphisms have been characterized in patient tumor samples from clinical trials using conatumumab for the treatment of soft tissue sarcoma [130], colorectal cancer [122,132], and NSCLC [125] and tigatuzumab for the treatment of NSCLC [129] in an effort to determine if FcγR polymorphisms segregated with outcomes. None of the FcγR polymorphisms were found to be significantly associated with PFS or overall survival (OS), but patients homozygous for the high affinity binding alleles who received conatumumab in combination with paclitaxel and carboplatin for NSCLC had a trend toward longer OS [125]. Further, there was a trend towards significance for improved PFS in patients with F158V polymorphisms when conatumumab was used in in combination with other agents [122]. Patients with high affinity binding alleles of FcγR may have experienced a marginally enhanced benefit from conatumumab and tigatuzumab. Modifying the Fc region to increase the affinity for the FcγR has been proposed as one strategy to enhance the potency of TRAIL DR antibody agonists in the clinical setting [134,141,142,143,144,145].

DR4 and DR5 levels were examined by immunohistochemistry (IHC) in tumor tissues [20,23,116,117,118,119,126,131,133,146,147,148], although the levels of DR4 and DR5 expression have not always correlated with response in the preclinical setting [30]. Expression levels varied tremendously within and between tumors and did not correlate with response to treatment [20,23,116,117,118,119,120,126,131,133,146,147,148,149].

Determining whether the TRAIL DR agonist has reached the tumor and initiated apoptosis has been challenging to assess in patients. Circulating DNA, active caspase-3, active caspase-8, and full-length and caspase-cleaved forms of the intermediate filament protein, cytokeratin 18 [150], have been utilized as biomarkers of apoptosis [151], and have been measured in clinical trials to evaluate the effectiveness of TRAIL DR agonists [19,20,22,122,126]. However, the levels of these biomarkers of apoptosis have varied among tumors and have not correlated with response. This is likely because apoptosis is a rapidly occurring event, and repeated measurements of apoptosis in tumors is difficult in clinical samples. Specimens that are available for assaying may not be representative of TRAIL DR agonist-induced tumor cell death as apoptotic cells are rapidly engulfed and destroyed by phagocytic cells in the surrounding microenvironment [151]. Considering the difficulty of obtaining solid tumor biopsies, acquiring serial samples of easily accessed tissues, such as blood, for the evaluation of tumor cell apoptosis by validated assays may provide the most helpful insights into TRAIL DR agonist activity in tumors [151,152]. The M30 and M65 sandwich enzyme-linked immunosorbent assay (ELISA) systems, for example, enable detection of caspase-cleaved and uncleaved (total) cytokeratin 18 (CK18) in serum, respectively [151,153,154,155]. In cells undergoing apoptosis, caspase cleaves CK18, and cleaved CK18 is released into the serum while total CK18 is released into the serum by cells undergoing necrosis. Serum samples are generally more easily obtained than serial tumor tissues for the assessment of apoptotic biomarkers. Thus, the M30 and M65 assay systems may provide more information about tumor cell death in real time than the IHC of apoptotic biomarkers in tumor tissues that are infrequently acquired. Finally, recent work has evaluated positron emission tomography imaging-based sensors of caspase-3 activation as a non-invasive means to measure apoptosis in tumors in real time [156,157,158,159]. Biomarkers of apoptosis that provide real-time information about the effectiveness of TRAIL DR agonists and therapeutic agents in general hold the potential to expediently identify sensitivity or resistance to treatment.

## 6. Strategies to Improve DR Agonists in Patients with Breast Cancer

The failure of TRAIL agonists in the clinic despite robust preclinical data raises questions about the reasons for the lack of activity. As outlined below, the need for predictive biomarkers, higher potency agonists with longer half-lives, and combination therapy all will need to be addressed for TRAIL agonists to successfully translate to the clinic.

### 6.1. Predictive Biomarkers

Patient selection criteria (predictive biomarkers) that will help identify who may benefit from TRAIL DR agonist therapy may provide the most critical starting point for enhancing TRAIL DR agonist performance in patients with cancer. To date, patients who have received TRAIL DR agonists have been unselected.

TNBC cell lines with a mesenchymal (also known as basal B) phenotype are extremely sensitive to apoptosis induced by TRAIL agonists, whereas cell lines representative of the other subtypes (ER+/luminal, HER2 amplified, and basal A TNBC) of breast cancer are comparatively resistant [30,93,94]. The mechanistic basis of the differential sensitivity of the different subtypes of breast cancer cell lines is not known. The basal B TNBC cell lines highly expressed the mesenchymal marker vimentin and expressed very little of the epithelial marker E-cadherin [30,93,94]. Recently, TNBC have been categorized into several subtypes by transcriptomic analysis [160,161]. The basal B TNBC cell lines that are most sensitive to TRAIL agonists are typically characterized as mesenchymal/mesenchymal stem-like based on these analyses and would represent approximately 20–25% of all TNBC [160,161]. On the other hand, E-cadherin was found to play an important role in promoting DR4 and DR5 clustering and formation of the Death-Inducing Signaling Complex (DISC) [120] in pancreatic, lung and colon cancer cell lines and higher E-cadherin expression was associated with a lower IC50 of TRAIL [120]. These contrasting findings also suggest that tissue-specific characteristics may need to be considered when identifying a biomarker for TRAIL DR agonists that is relevant to any specific tumor type. Nonetheless, the data in TNBC suggest that selecting patients based on mesenchymal features (e.g., vimentin expression or mRNA expression-based classification) may enrich for patients likely to respond to TRAIL agonists.

Other predictive biomarkers have been characterized and proposed in previous preclinical studies. O-glycosylation was reported to be necessary for full activity of TRAIL DRs in multiple cancer cell lines [162]. Higher expression of O-glycosylation genes, specifically *GALNT14* in NSCLC, melanoma, and pancreatic cancer cell lines and *GALNT3*, *FUT3*, and *FUT6* in colon cancer cell lines, were associated with sensitivity to TRAIL-induced apoptosis [162]. Expression levels of O-glycosylation genes served as better predictors of response to TRAIL than the anti-apoptotic proteins BCL-2, XIAP, or FLIP. DR5 is O-glycosylated by GALNT14, and O-glycosylation of DR5 enhanced ligand–receptor clustering to facilitate more efficient downstream signaling. Overexpression of specific O-glycosylation genes enhanced TRAIL-induced apoptosis in pancreatic cancer and NSCLC cells, whereas inhibition of specific O-glycosylation genes attenuated TRAIL-induced apoptosis in melanoma and pancreatic cancer cell lines. Therefore, evaluation of expression levels of O-glycosylation genes has been proposed as a strategy to predict sensitivity to TRAIL, and efforts have been directed toward the development of a clinically relevant IHC assay [22]. Inhibition of syndecan-1 was associated with enhanced sensitivity to TRAIL due to TRAIL DR O-glycosylation in myeloma cells [163]. In a phase 2 study combining dulanermin with FOLFOX6 and bevacizumab for colorectal cancer, high expression of GALNT14 evaluated by IHC has been found to significantly associate with PFS and OS in the treatment over the control arm, suggesting GALNT14 expression served as a predictor of sensitivity to cytotoxic agents [22]. Expression of O-glycosylation genes, however, has not consistently been associated with TRAIL sensitivity in preclinical studies as mentioned above. In TRAIL-sensitive TNBC cells, elevated expression of O-glycosylation genes was not observed [30], suggesting that biomarkers for TRAIL resistance and sensitivity may need to be determined on a tumor-specific basis.

In head and neck cancer cell lines, caspase-8 and Bid expression has been shown to be associated with sensitivity to TRAIL [164]. Caspase-mutations occur in ~9% of head and neck cancers and have been found to confer resistance to TRAIL-induced apoptosis in head and neck cancer cell lines, as well as promote cancer cell migration, invasion, and tumor proliferation [164]. Measuring expression and genotyping caspase-8 in head and neck cancers may be predictive of TRAIL DR agonist efficacy. Caspase-8 has been evaluated by IHC of tumor sections in clinical trial of pediatric solid tumors [116]; however, caspase-8 expression was not associated with response. Others have shown that caspase-8 expression does not predict sensitivity to TRAIL in breast cancer or TNBC [30]. This suggests that caspase-8 expression or mutation may be a useful biomarker in head and neck cancer, but not across all tumor types including TNBC.

The data to date suggest the best biomarker for TNBC selection is the mesenchymal phenotype which can be identified either by assessing vimentin staining [93] or by a more comprehensive transcriptome analysis to identify the mesenchymal TNBC [160,161]. Further characterization of the sensitivity or resistance to TRAIL in breast cancer cells is needed to identify a mechanistic basis for the selectivity seen in preclinical studies as this will likely lead to a more robust predictive biomarker.

### 6.2. TRAIL and Anti-DR5 Agonist Antibody Combination Therapy

Increasing the potency of TRAIL DR agonists has been proposed as a more effective therapy [40,165,166]. The combination of two different TRAIL DR agonists has been explored, specifically TRAIL and the DR5-specific antibody conatumumab [40]. Crosslinking is required to mediate conatumumab activity, and findings from a clinical trial suggest that patients with a high-affinity FcγR-binding allele experienced longer OS with conatumumab treatment [20]. However, the combination of TRAIL and conatumumab overcame the requirement for crosslinking while enhancing DISC formation, downstream signaling, and induction of apoptosis in preclinical models including TNBC models [40]. This combinatorial strategy demonstrates a novel system for overcoming a possible lack of crosslinking that may occur in vivo. However, co-treatment with conatumumab and TRAIL did not supersede resistance to apoptosis in all cancer cells, again suggesting that tissue-specific considerations must be taken into account, and the addition of TRAIL to other TRAIL DR antibody agonists did not overcome the requirement for additional crosslinking in all combinations [40]. Although this strategy has not yet been tested in clinical trials.

### 6.3. High-Potency TRAIL Receptor Agonists

Highly potent TRAIL DR agonists (termed super agonists) have also been developed to attempt to improve activity, including DR5-specific multivalent scaffold proteins and Nanobody^®^ superagonists. The multivalent scaffold proteins are based on the third fibronectin type III domain of the glycoprotein tenascin C, which possesses a region similar to the variable region characteristic of antibodies [93,165]. The antibody variable-like region can undergo mutation that will allow for interaction with different proteins. The DR5-specific multivalent scaffold proteins were found to induce apoptosis at picomolar concentrations (compared to nanomolar potency for TRAIL), and enhanced apoptosis was observed with increased valency of the super agonist [165]. When one of these super agonists (MEDI3039) was tested in breast cancer cell lines, the agent showed remarkable activity with IC50s in the picomolar range, again with the mesenchymal TNBC lines most sensitive [94]. MEDI3039 showed dramatic effects in orthotopic xenografts and metastatic models treating early metastases or established metastases [94]. Pharmacokinetics of MEDI3039 in mice (t1/2 = 15.4 h; Cmax = 1–2 µM) and in cynomolgus monkeys (t1/2 = 31 h) suggest that drug levels could be above the predicted IC50 for the mesenchymal TNBC for a week or longer after a dose [94]. Despite the remarkable efficacy of MEDI3039 in TNBC preclinical models, the effectiveness of this multivalent scaffold protein super agonists is yet to undergo testing in clinical trial.

Similarly, a multivalent DR5-specific Nanobodies^®^, which is based on single variable domains derived from camelid heavy chain antibodies and linked together to form multivalent proteins, more potently and rapidly induced apoptosis than TRAIL or the DR5-specific antibody LBY135 [166]. As mentioned above, unexpected hepatotoxicity and immunogenicity were observed in a phase I clinical trial using TAS266, a DR5-specific agonist Nanobody^®^, that resulted in early cancellation of the trial [113]. These findings have raised concern about the further development of Nanobodies^®^ for TRAIL pathway activation for cancer treatment. There are ongoing phase I and II trials for a tetravalent DR5 agonistic antibody and a multimeric anti-DR5 IgM agonist antibody (NCT04950075, NCT03715933, NCT04553692) whose efficacy and safety are still to be determined (Table 1).

Such high potency TRAIL death receptor agonists are likely to have better exposure of the tumors in vivo to therapeutic levels of the drug so that testing these high-potency agonists in TNBC will be of interest. However, as described above, patient selection may remain a major criteria required for TRAIL agonist efficacy as the higher-potency agonists (e.g., MEDI3039) failed to overcome resistance [94].

### 6.4. Combination Therapy with Non-Targeted and Targeted Agents in Breast Cancer

The identification of synergistic combinations with TRAIL DR agonists has been proposed as a strategy to overcome resistance to TRAIL and has already been widely explored in preclinical contexts [90,109,159,164,167,168,169]. The combining of drugs has been proposed to overcome resistance to cancer therapies by reducing variability among cells with respect to timing and susceptibility to an apoptosis inducer [170,171]. Moreover, TRAIL DR agonists are able to activate apoptosis independently of p53 which is commonly mutated in triple negative cancer and this characteristic provides another rationale for using TRAIL DR agonists in combination with other therapeutic agents to overcome resistance to cell death induction [46].

Chemotherapy has been found in preclinical studies to synergize with TRAIL in breast cancer cells, including doxorubicin, cisplatin, 5 fluorouracil, gemcitabine, and irinotecan, and paclitaxel [30,172]. The mechanisms proposed to sensitize cells to TRAIL include enhanced DISC formation, upregulation of pro-apoptotic regulators (e.g., TRAIL DRs), downregulation of anti-apoptotic regulators, and independent synergistic caspase activation by the chemotherapy. In a xenograft mice TNBC model and TNBC patient derived metastatic cells, etoposide along with TRAIL and doxorubicin enhanced cellular apoptosis with concomitant increase of DR5 [173]. YM155, which is a small molecule selective survivin inhibitor, in association with CD34+ TRAIL engineered cells was reported to increase cell cytotoxicity and apoptotic response to TRAIL by upregulation of DR5 in three TNBC cell line models [28].

Inhibition of anti-apoptotic components of the intrinsic pathway, including BCL-2 family members [55] and Inhibitor of Apoptosis Proteins (IAPs) [157,174], have been shown to sensitize cancer cells to TRAIL in TNBC models [175,176,177,178,179,180]. Inhibition of another anti-apoptotic BCL-2 family member, BCL-xL, has been shown to increase TRAIL-mediated apoptosis in breast cancer cells including TNBC in preclinical models [55]. Mechanistically, the data suggest that inhibition of anti-apoptotic BCL-2 family members either genetically or pharmacologically results in increase DR-induced activation of caspase-9 and activation of caspases-3 and -7. Further, activation of caspases-3 and -7 by caspase-9 results in retrograde activation of caspase-8. Thus, in the setting of BCL-2 family inhibition, the direct DR-induced activation of caspase-8 induces a cascade of downstream caspases including caspases-3, -7 and -9 which in a retrograde manner activate caspase-8 thus creating a reinforcement loop resulting in more apoptosis. Preclinical data suggest that IAP-inhibiting SMAC mimetics can enhance TRAIL mediated apoptosis in breast cancer models [157]. IAP inhibitors target IAP1, IAP2 and XIAP for degradation leading to increased activation of caspases in breast cancer cells [174]. Combining the SMAC mimetic birinapant and TRAIL has shown promising activity in TRAIL-resistant breast cancer by down- regulating cFLIP [181].

Several natural products have been also investigated in combination with TRAIL in different types of cancers [182]. For TNBC, piperine, an alkaloid, was reported to enhance TRAIL mediated TNBC tumor killing both in vitro and in vivo by suppressing survivin and decreasing NF-κB pathway activity [183,184]. These data are supported by studies overexpressing a nondegradable version of the NF-κB inhibitor (IκBΔN) which prevents NF-κB activation and potentiates TRAIL induced apoptosis in breast cancer cells [58]. Similarly, pterostilbene, a naturally occurring resveratrol analogue, enhanced TRAIL mediated apoptosis in TNBC cell lines by decreasing the levels of anti-apoptotic proteins [185].

Histone deacetylase inhibitors (HDACi), which alter gene expression by modifying chromatin structure by preventing the removal of acetyl groups from histone tails, have been found to synergize with TRAIL. Vorinostat, an HDACi, along with TRAIL increased anoikis by causing dissipation of mitochondrial membrane potential, activation of caspase-3 and reduction of phosphorylation of BCL-2 family protein in breast cancer cells [186]. Chidamide, another HDACi, enhanced TRAIL-induction of apoptosis in breast cancer models by increasing autophagy [187]. As one of the HDAC inhibitor-mediated mechanisms of TRAIL sensitization, FLIP downregulation and DR5 upregulation have been documented in other systems. [188,189].

In studies in breast cancer cells, inhibition of the cell cycle G2M checkpoint regulator Wee1 increased TRAIL mediated apoptosis in TNBC in part by upregulating DRs on the breast cancer cells [190]. Interestingly, CDK9 inhibition has also been shown to increase TRAIL mediated apoptosis in other cancer models by downregulating the anti-apoptotic regulators MCL-1 and FLIP, thus indicating that inhibition of CDK9 sensitizes cells to TRAIL through regulation of components of the extrinsic and intrinsic pathway [172]. However, CDK9 inhibition has not been tested in breast cancer models. Inhibition of the epidermal growth factor receptor (EGFR) as well as receptor tyrosine kinase (RTK) has been shown to increase TRAIL receptor mediated apoptosis in mesenchymal TNBC cells but not in the non-mesenchymal TNBC [30]. The mechanistic basis of this is not determined but EGFR can activate both proliferative (e.g., MAPK) and anti-apoptotic (e.g., AKT) pathways [191]. Interestingly, in HER2 amplified breast cancer cells, inhibition of HER2 increased TRAIL receptor agonist mediated apoptosis and mechanistically this was primarily due to the inhibition of AKT signaling [167] suggesting that the effects of EGFR inhibition may be similar. Activation of MET, another RTKs that frequently expressed in TNBC has also been shown to induce resistance to TRAIL-mediated apoptosis in several models including breast cancer [192]. Again, the AKT pathway appeared critical to this suggesting that MET inhibition should be explored to potentiate TRAIL death receptor induced apoptosis.

Several studies have shown that radiation can sensitize breast cancer cells to TRAIL death receptor agonists [89,92,193]. In the paper by Chinnaiyan et al., they found that in different breast cancer cells including TNBCs, the TRAIL potentiating effects of radiation were mediated by increased DR5 expression and depended on p53 expression [92]. However, Buchsbaum et al. demonstrated that the combination of a TRAIL death receptor agonist and radiation or TRAIL agonist, doxorubicin, and radiation showed increased tumor growth inhibition compared to single agents in an in vivo xenograft model using a derivative of the MDA-MB-231 cell line which is p53 mutant [89]. In another study, the investigators overexpressed TRAIL and the second mitochondrial activator of caspases (SMAC) in the MDA-MB-231 cell line and showed that overexpression of TRAIL or SMAC alone increased radiation sensitivity and the combination of all three had the most efficacy. Their data suggest that the synergy is mostly driven by increased activation of the mitochondrial apoptosis pathway. While these data are intriguing, breast cancer is a systemic disease; the role of combining TRAIL death receptor agonists would need to be carefully considered in instances where increased local control would be beneficial, such as in the setting of locally advanced or inflammatory breast cancer. Immune therapies have shown efficacy in TNBC [2,3,4,5,6]. One recent study has shown TRAIL-induced transcriptional upregulation of PD-L1 expression in the MDA-MB-231 TNBC cell line via an ERK-dependent mechanism [194]. Interestingly, RNAi-mediated knockdown of PD-L1 increased sensitivity to TRAIL agonists in vitro, suggesting a non-immune-mediated mechanism of resistance mediated by PD-L1. A recent paper describes a phase I clinical trial in ovarian cancer where autologous monocytes were elutriated from patients, mixed with interferons and then infused intraperitoneally showing safety and some activity [195]. In a preclinical work described in the paper, the investigators demonstrated that the activity of the interferon-treated monocytes was mediated by induction of TRAIL on the monocytes. While not tested by systemic delivery or in TNBC models or patients, this raises the possibility of using autologous cells to deliver TRAIL in TNBC.

Further studies are necessary to determine the optimal drug combinations required to sensitize TNBC cells to DR agonists.

### 6.5. Mesenchymal Stem Cell-Mediated TRAIL Delivery

Mesenchymal stem cells (MSCs) which possess tumor-homing capabilities and can evade elimination by the immune system have been engineered as gene therapy delivery systems and have been explored as TRAIL delivery agents [196,197]. Mesenchymal stem cells engineered to express TRAIL (MSC-TRAIL) induce apoptosis more potently than soluble TRAIL [198] and have been found to induce apoptosis in sarcomas, tongue squamous cell carcinoma, glioblastoma, and TNBC brain metastases in vitro and in vivo [199,200,201,202]. In another TNBC brain metastasis model, exosomes isolated from MSC expressing the CXCR4 chemokine receptor and TRAIL potentiated the antitumor efficacy of TRAIL [203]. The use of MSC to target TNBC brain metastasis is novel and may overcome the issues of penetration of the blood–brain barrier for TRAIL death receptor agonists. Cisplatin was found to sensitize mouse glioblastoma tumors to MSC-delivered TRAIL in vitro and in vivo [198], suggesting that combinatorial therapies may effectively sensitize cancer cells to stem cell-delivered TRAIL. Similarly, MSC-TRAIL combined with an AMPK inhibitor enhanced killing of glioblastoma cells by increasing expression of the pro-apoptotic BCL-2 family protein Bax and simultaneously reducing the expression of the anti-apoptotic proteins FLIP, XIAP and BCL-2 [204]. Adipose tissue derived stem cells (ASCs) can induce cancer cell death by secretion of TRAIL and Type I interferons. In a colon cancer study TRAIL expressing ASCs induced by M1 macrophages decrease the level of CD133+ cancer stem cells and decreased the number of M2 macrophages, thus boosting the anti-tumor efficacy of TRAIL [205]. Exosomes secreted from MSCs are promising biological tools for targeted therapy in cancer. In this context, the injection of exosomes armed with TRAIL (Exo-TRAIL) secreted from MSCs reduced the tumor volume in tumor-bearing mice [206]. In a separate study using TRAIL resistant MCF-7 ER+ breast cancer cells, cell death was significantly increased when treated with ASCs overexpressing SMAC and TRAIL [207]. MSCs transduced with TRAIL induced apoptosis in cancer cells in vitro, and reduced tumor growth and lung metastasis in a TNBC model in vivo [208]. MSC-TRAIL are currently being tested in patients with lung cancer as phase1/2 trial (NCT03298763) (Table 1). Whether this delivery system overcomes the lack of activity with soluble TRAIL agonists remains to be determined.

### 6.6. Nanoparticle Mediated TRAIL Delivery

Recent advances in drug delivery using nanotechnology are now being used to develop TRAIL encapsulated nanoparticles as a safe and an effective delivery method. As described above, major shortcomings of soluble TRAIL therapeutics may be attributed to a short half-life of rhTRAIL and poor penetration of the tumor stromal matrix [209]. Modified therapeutics with materials such as polyethylene glycols (PEG) or serum proteins have been shown to enhance the half- life in circulation along with improved target- site delivery and limited off target effects [182,210]. A PEGylated human recombinant TRAIL named TLY012 ameliorated skin fibrosis, pancreatic disease and cirrhosis in preclinical models [211] and is being actively investigated for anticancer effects in tumor models but is yet to be tested in TNBC models [212]. Recently, a nanogel delivering nitric oxide and TRAIL to pancreatic ductal adeno carcinoma xenografts had enhanced tumor penetration and efficacy [213]. A paclitaxel (PTX)-bound albumin nanoparticles with embedded TRAIL (TRAIL/PTX HSA-NP) formulation based on paclitaxel-bound albumin nanoparticles embedded with TRAIL showed improved TRAIL efficacy both in vitro and in vivo pancreatic cancer models [214]. Curcumin which is known to have synergy with soluble TRAIL has also been used in combination with TRAIL in nanoparticles to potentiate TRAIL-induced apoptosis in various cancer models including prostate cancer, brain tumors, and in colon cancer [215,216,217]. A membrane targeted protein-based nanoparticle with the dipeptide, diphenylalanine, and TRAIL was shown to potentiate apoptotic death in TRAIL-resistant MCF-7 breast cancer cells [218]. TRAIL-coated gold nanoparticles enhanced TRAIL killing activity in NSCLCs by a mechanism involving mitochondrial fragmentation and autophagy [219]. In a colorectal cancer derived xenograft model, TRAIL-iron oxide nanoparticles induced greater apoptosis when compared to soluble TRAIL monotherapy [220]. Use of TRAIL delivery by means of ferric oxide nanoparticles combined with bortezomib and silver nanoparticles has also reported [221,222]. A recent study used a neutrophil membrane-based nanoparticle delivery system to create nanoparticles loaded with paclitaxel and TRAIL that demonstrated anti-tumor efficacy [223]. Interestingly, combining curcumin-loaded chitosan nanoparticles with placental derived mesenchymal stem cells (PDMSCs) expressing TRAIL augmented TRAIL induced apoptotic killing in TNBC [224]. Similarly, hyaluronic acid-coated and pH-sensitive polymeric nanoparticles containing the drug embelin along with TRAIL-enhanced cytotoxic and apoptotic effects against TNBC cells [225].

Altogether, nanoparticle-mediated TRAIL delivery may enhance efficacy and merits further study in TNBC.

### 6.7. Immunomodulatory Effects of TRAIL Agonists

In addition to the direct anti-tumor apoptotic effect there is accumulating evidence of a complex interplay between TRAIL and tumor immune microenvironment. Tumor microenvironment contains heterogenous cell populations and TRAIL-R agonists have been shown to modulate the function of different immune cells (Figure 2). The TRAIL/TRAIL-R system is known to exert both pro- and anti-tumor activities via immune modulation involving both endogenous and exogenous TRAIL signaling in the tumor microenvironment [42,226,227,228]. As demonstrated in Figure 2, the antitumorigenic immune components of tumor microenvironment are natural killer (NK) cells, dendritic cells (DC), cytotoxic T lymphocytes (CTLs), and M1 tumor-associated macrophages (M1 TAMS) secreting mainly proinflammatory cytokines like IL-1β, IL-1α, IL-6, IFNγ, TNFα etc. [229,230,231]. The pro-tumorigenic components usually involve secretion of anti- inflammatory cytokines derived either from myeloid derived suppressor cells (MDSCs), regulatory T cells (T-regs), cancer-associated fibroblasts (CAFs), B cells, and tumor-associated neutrophils (TANS) [232,233,234]. Neutrophils can secrete neutrophil extracellular traps (NETs), which are structures that consist mainly of DNA fibers and antibacterial proteins. Besides playing a role as a host defense system, NETs are known to have pro-metastatic potential and impact immune surveillance in the tumor microenvironment [235].

NK cells are a major effector cell of the innate immune system that can eliminate tumor cells either by degranulation (granzyme) or by expressing Fas Ligand (CD95L) or TRAIL on their surface which in turn interacts with DRs on the tumor cells to induce apoptosis [236]. In a syngeneic mouse renal adenocarcinoma tumor model therapeutic IL-12 induced production of IFNγ by the NK cells led to upregulated TRAIL surface expression on NK cells and substantially suppressed tumor metastasis in lungs and liver [237,238].

CTLs are a major anti-tumor component of the of adaptive immune system destroying the tumor cell targets by similar mechanism as NK cells [239,240,241]. The expression of TRAIL on CTLs is activated by interactions of the CTL with the tumor and tumor-produced cytokines [227]. For example, autologous lung cancer cells induce TRAIL on tumor-infiltrating CD4+ CTLs contributing to tumor specific CD4+ CTLs mediated cell death [242]. Reverse signaling by TRAIL has also been reported in mouse T-cells where binding to the DRs induced TRAIL-mediated activation of p38 MAPK resulting in proliferation of the T cells and increased production of IL-2, IL-4, and IFNγ by the T cells [85].

Dendritic cells act as a heterogenous population of immune cells that links both innate and adaptive immune system [243]. Intriguingly, an engineered conditionally replicating adenovirus (CRAD5) encoding the TRAIL protein (CRAD5-TRAIL) has been reported to show promising antitumor efficacy in several murine tumor models [244]. Recently CD40 ligand (mCD40L) in combination with CRAD-TRAIL induced immune activation of dendritic cells DCs, B cells, and tumor-infiltrating T cells along with upregulation of DRs on the tumors thus elucidating a novel strategy for oncolytic adenovirus-mediated solid tumor immunotherapy [244,245].

Neutrophils are another major component of the innate immune system and NETosis is an immune response produced by cytokine activated neutrophils [235]. A recent report describes the engineering of neutrophils to express an eGFP-TRAIL which is highly positively charged and binds to the negatively charged DNA in the NETs. These eGFP-TRAIL NETs trap tumor cells and induce apoptosis by activation of the DRs on the tumor and retain the anti-microbial property of the NETs [246].

While not yet studied in breast cancer, TRAIL may also have pro-tumorigenic effects on immune cells. TRAIL can potentiates the activity of Treg cells which are well-known contributors to immune escape of tumors [247]. Tregs secrete various anti-inflammatory cytokines such as TGFβ, IL-10, and IL-35 which inhibits anti-tumor immunity by suppressing CTL activity and antigen presentation by DCs [248]. Upregulation of TRAIL in Tregs resulted in DR5-dependent cytotoxicity against CD4+ T cell [249]. Another pro-tumorigenic role for TRAIL was demonstrated in a TRAIL-resistant syngeneic murine orthotopic pancreatic cancer model where tumors grew faster, and survival was worse in wild-type mice compared to TRAIL null mice [250]. TRAIL treatment of wild-type mice enhanced tumor growth and increased the number of CD4+ Tregs in the tumors [250]. Colorectal cancer tumor derived extracellular vesicles (TEV) can secrete an isoform of DR5 which can prevent TRAIL-induced apoptosis. Similarly, these TEVs bearing TRAIL have also been reported to induce T-cell apoptosis in vitro and in vivo thus supporting a role of TRAIL in tumor progression [251,252]. A novel splice variant of TRAIL named TRAIL short found in TEVs acts as a dominant negative ligand which may confer resistance in TRAIL-sensitive tumor and immune cells [253].

Despite having majorly antitumorigenic effect, NK cells may also have an impact on tumorigenesis [254]. A study reported that meningeal IFNγ+ NK cells produce interferon which modulates the activity of inflammation by a novel subset of astrocytes that are with LAMP1+ TRAIL+ phenotype, which in turn bind to the TRAIL DRs on activated T-cells to suppress the inflammatory response in murine autoimmune encephalomyelitis model [255]. This suggests that the meningeal NK cells could promote CNS metastasis by suppressing inflammation. In another study, a second potential pro-tumorigenic role of TRAIL expressing NK cells was described where the NK cells impaired the generation of MHC class I antigen peptide complexes on dendritic cells (DCs) by activation of the DRs on the DCs [238]. This resulted in impaired cross priming of CD8+ T-cells, thus ultimately inhibiting the CD8+ T cell anti-tumor response [256].

Neutrophils and macrophages also contributes to the pro-tumorigenic activity of TRAIL where tumor secreted cytokines act as mediators of immune suppression [227]. An in vivo study in murine hepatocarcinoma and melanoma models found that IL-35 induces a conversion of neutrophils from N1 (anti-tumorigenic) to an N2 (pro-tumorigenic) state. This was accompanied by a decrease in level of TRAIL expression in transformed neutrophils consistent with their inability to kill the tumor cells [257].

Inflammatory cytokine secretion in tumor micro-environment can elevate chronic inflammation resulting in immune suppression [258]. Cytokines such as IL-8, TNF-α, CCL20, MIP-2 and MIP-1β have been shown to be dramatically induced by activation of TRAIL-R1 and R2 in different tumor cell lines in an NF-kB dependent manner [228,259,260,261]. Interestingly induction of the proinflammatory cytokines was independent of caspase activity, however caspase-8 protein was essential as a scaffold for cytokine induction [262]. Notably, CCL2 was identified as the main cytokine controlling the formation of tumor-supportive compartment by both recruitment of tumor supportive myeloid cells and polarizing monocytes into an M2 macrophage phenotype. These data suggest that blocking cytokine induction by TRAIL or blocking cytokine function may be a therapeutic strategy to inhibit the pro-tumorigenic effects of TRAIL and improve TRAIL based therapy [228].

Although breast cancer has been regarded as immunologically “cold”, clinical studies and new drugs in the present decade have shown immune modulation-based therapies as potential regimens in treating breast cancer patients. Anti-PD1/PDL1immune checkpoint inhibitors have efficacy in TNBC and pembrolizumab has been FDA-approved in both the neo/adjuvant and metastatic setting [3,4,5,6]. Other T-cell-targeted immune therapies are being evaluated in TNBC such as targeting cytotoxic T-lymphocyte associated antigen-4 (CTLA-4) [263], and chimeric antigen receptor-modified T cells (CAR-T) [264,265]. A variety of vaccine therapies are also under investigation supporting the immune modulatory autologous system against breast cancer [266]. Interestingly, tumor nano lysates (TNLs) derived from cancer cells as a result of TRAIL and fluid sheer stress are under investigation in a pilot in vivo study as a preventive vaccine for TNBC [267].

To date, no studies of the immune modulatory effects of TRAIL have been reported in breast cancer models. Hence, understanding how TRAIL modulates immune milieu may allow selection of the optimal combination of immune therapies in TNBC.

## 7. Conclusions and Future Directions

The TRAIL apoptotic pathway has been an enticing therapeutic target for the treatment of TNBC. Because TRAIL has been well-characterized as an inducer of apoptosis selectively in cancer cells with minimal toxicity to normal cells, DR4 and DR5 agonists have been tested in phase 1, 2, and 3 clinical trials. Despite a robust and selective killing effect in many preclinical models including TNBC in vitro and in vivo, TRAIL DR agonists, though well-tolerated, has not improved outcomes in patients with advanced solid tumors and hematological malignancies. This conundrum poses several challenges to improving the effectiveness of TRAIL DR agonists in the clinic. The plausible strategies to enhance the potency of TRAIL in TNBC are illustrated schematically in Figure 3.

First, improving the pharmacokinetic features (e.g., potency, half-life) and delivery of TRAIL agonists (e.g., via nanoparticles or MSCs) will be needed to have optimal efficacy in TNBC and other tumors. Second improving the assessment of the pharmacodynamic effects of TRAIL DR agonists would provide better insight into whether the drugs are able to induce apoptosis in tumors especially in solid tumors such as TNBC. Third, identifying biomarkers that predict response may aid in selecting patients who will benefit the most from TRAIL DR therapy such as the selection of patients with mesenchymal TNBC. However, biomarkers may be disease-specific, as findings have produced conflicting results concerning biomarkers of TRAIL sensitivity. Markers of epithelial–mesenchymal transition (EMT) have been associated with enhanced sensitivity to TRAIL in TNBC, whereas markers of EMT have been associated with resistance in pancreatic, colon, and lung cancer cells [30,120]. Fourth, combining TRAIL DR agonist therapies with other chemotherapeutic and targeted agents may effectively enhance treatment by limiting the inherent variability observed among cancer cells [170,171]. Although efforts toward combining TRAIL DR agonists and other agents have been carried out in clinical trials, the results from those studies have not identified a combinatorial therapeutic strategy that improved by the addition of a TRAIL DR agonist. Efforts toward further characterizing regulation of the TRAIL pathway in TNBC may provide crucial insights into understanding the molecular underpinnings that control sensitivity to TRAIL [55] and guide the development of novel therapeutic agents or drug combinations to enhance TRAIL DR agonist effectiveness in patients with TNBC. Finally, TRAIL has both pro-tumorigenic and anti-tumorigenic effects on the immune milieu. Identifying the critical determinants of each will likely allow combination of TRAIL agonists with immune modulatory drugs for improved efficacy in TNBC.

TRAIL DR agonists are still promising therapeutic agents for inducing apoptosis in TNBC. Improving strategies to promote the effectiveness of TRAIL DR agonists in the clinical setting are warranted.

## Figures and Tables

**Figure 1 cells-11-03717-f001:**
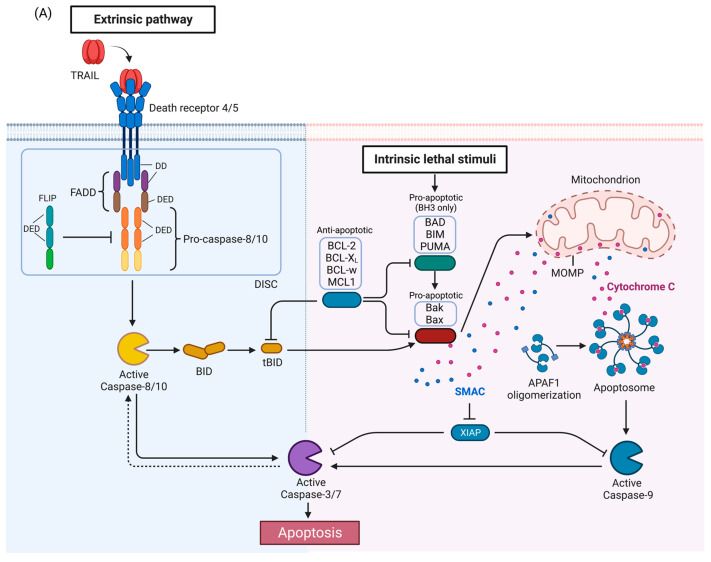
TRAIL signaling pathway. (**A**) Apoptosis pathway. Activation of DR4 and DR5 by TRAIL induces the extrinsic apoptosis pathway (left). The intrinsic pathway (right) is activated by a variety of stimuli and leads to release of proapoptotic proteins from the mitochondria. The two pathways interact as caspase-8 activated by the DRs can cleave BID which then activates the intrinsic pathway, and conversely, caspase-3 can cleave and activate caspase-8 in a feedback loop, thus amplifying the apoptotic signal. (**B**) TRAIL-mediated NF-kB/MAPK pathways. Anti- or pro-survival mechanisms appear to be context dependent. (**C**) TRAIL-mediated necroptosis pathway. (**D**) Crosstalk between TRAIL pathway and autophagy. APAF1, Apoptotic protease activating factor 1; DD, death domain; DED, death effector domain; DISC, death-inducing signaling complex; FADD, Fas-associated death domain protein; FLIP (FLICE [FADD-like IL-1β-converting enzyme]-inhibitory protein); MOMP, mitochondrial outer membrane permeabilization; SMAC, second mitochondrial activator of caspases; XIAP, X-linked inhibitor of apoptosis. TRAF2, TNF receptor-associated factor 2; NEMO, NF-kappa-B essential modulator; TRADD, Tumor necrosis factor receptor type 1-associated DEATH domain protein; NF-κB, nuclear factor-κB; MAPK, mitogen-activated protein kinase; MLKL, mixed-lineage kinase domain-like protein; ATG, Autophagy-related protein; LC3, Microtubule-associated proteins 1A/1B light chain 3; IAP: inhibitor of apoptosis. Figures were created with BioRender.com (accessed on 8 November 2022).

**Figure 2 cells-11-03717-f002:**
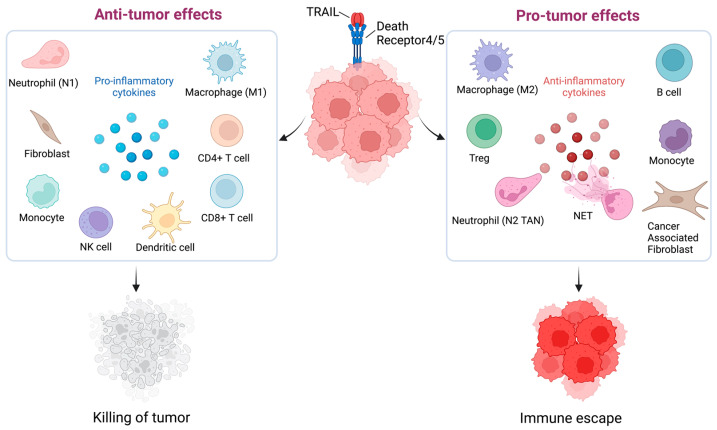
Immunomodulatory effects of TRAIL DR pathways. The TRAIL/TRAIL-R pathway can impart both pro- and anti- tumorigenic effect on the tumor microenvironment. Endogenous TRAIL expressed by the immune cells of the tumor microenvironment (left panel), or exogenous TRAIL contribute to the anti-tumorigenic defense mechanism of TRAIL, thus killing the tumor cells (shown on the left). On the other hand, exogenous TRAIL either directly or via immune cells may activate the secretion of inflammatory cytokines promoting infiltration of immune suppressor cells (M2 macrophages, Tregs, Tumor Associated Neutrophils [TAN]/N2 TAN, pro-tumoral monocytes and B cells), resulting in immune escape contributing to pro-tumorigenic effect of TRAIL (shown on the right). Figure was created with BioRender.com (accessed on 8 November 2022).

**Figure 3 cells-11-03717-f003:**
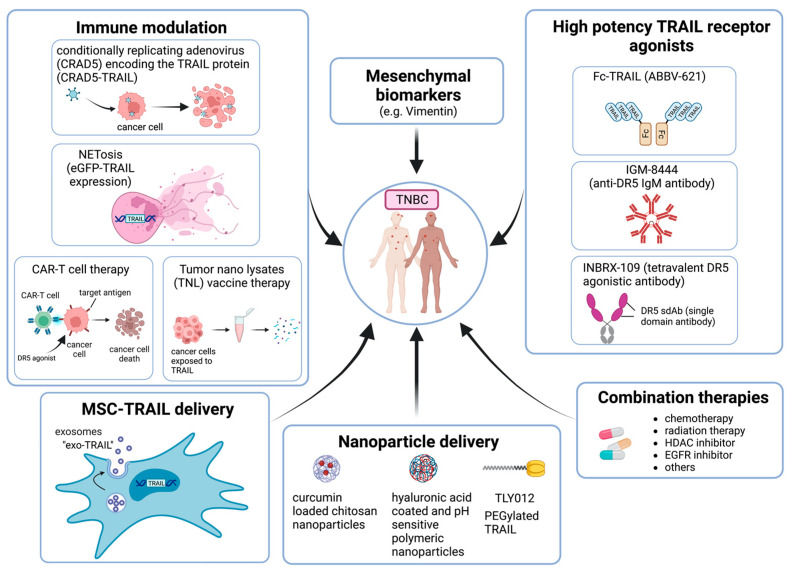
Plausible strategies to improve TRAIL efficacy in TNBC. Predictive biomarkers like Vimentin, high-potency TRAIL receptor agonists such as multivalent anti-TRAIL receptors antibody, different combinatorial therapies including chemotherapy, radiation therapy, HDAC inhibitors, EGFR inhibitors, different forms of nanostructures delivery system, MSC TRAIL delivery via exosome and along with possible immune therapies may enhance future TRAIL therapy in TNBC. Though to date, specific immune modulatory therapy has not yet been reported but there remains a scope to have the above illustrated immune modulatory delivery system to enter future clinical trials for TNBC. Figure was created with BioRender.com (accessed on 8 November 2022).

**Table 1 cells-11-03717-t001:** Summary of clinical trial for TRAIL/DR Pathway targeting therapy.

Molecular Type	Drug	Status	Study Title	Conditions	Interventions	**Phase**	**NCT Number**
rhTRAIL	Dulanermin	Terminated	A Study of Dulanermin in Combination with Rituximab in Subjects with Follicular and Other Low Grade, CD20+, Non-Hodgkin’s Lymphomas	Non-Hodgkin’s Lymphoma	Drug: Dulanermin	Phase 1/2	NCT00400764
					Drug: Rituximab		
	Dulanermin	Completed	A Study of Dulanermin Administered in Combination with Camptosar^®^/Erbitux^®^ Chemotherapy or FOLFIRI (with or without Bevacizumab) in Subjects with Previously Treated Metastatic Colorectal Cancer	Colorectal Cancer	Drug: FOLFIRI regimen	Phase 1	NCT00671372
					Drug: Bevacizumab		
					Drug: Cetuximab		
					Drug: Dulanermin		
					Drug: Irinotecan		
	Dulanermin	Completed	A Study of Dulanermin Administered in Combination with the FOLFOX Regimen and Bevacizumab in Patients with Previously Untreated, Locally Advanced, Recurrent, or Metastatic Colorectal Cancer	Metastatic Colorectal Cancer	Drug: FOLFOX regimen	Phase 1	NCT00873756
					Drug: Bevacizumab		
					Drug: Dulanermin		
	Dulanermin	Unknown	A Phase III Trial of Recombinant Human Apo-2 Ligand for Injection	Non-small-cell Lung Cancer (NSCLC) Stage IV	Biological: Recombinant human Apo-2 ligand for Injection	Phase 3	NCT03083743
					Biological: Placebo		
	Dulanermin	Completed	A Study of AMG 951 [rhApo2L/TRAIL] in Subjects with Previously Untreated Non-Small Cell Lung Cancer (NSCLC) Treated with Chemotherapy +/− Bevacizumab	Non-Small Cell Lung Cancer	Drug: AMG 951 (rhApo2L/TRAIL)	Phase 2	NCT00508625
					Drug: Bevacizumab		
					Drug: Carboplatin		
					Drug: Paclitaxel		
	Lexatumumab	Terminated	HGS-ETR2 to Treat Children with Solid Tumors	Ewing’s Sarcoma	Drug: Lexatumumab alone	Phase 1	NCT00428272
				Osteosarcoma	Drug: Lexatumumab in combination		
				Neuroblastoma	Drug: Interferon gamma 1b in combination		
				Rhabdomyosarcoma	Drug: Gamma 1b potential expansion		
	Circularly Permuted TRAIL(CPT)	Recruiting	A Multicenter, Randomized, Double-Blind, Controlled Phase III Study of CPT or Placebo in Combination with Thalidomide and Dexamethasone in Subjects with Relapsed or Refractory Multiple Myeloma	Multiple Myeloma	CPT+/−TD	Phase 3	ChiCTR-IPR-15006024
Recombinant human TRAIL-Trimer™ fusion protein	SCB-313	Completed	A Phase I Study Evaluating SCB-313(Recombinant Human TRAIL-Trimer Fusion Protein) for the Treatment of Malignant Pleural Effusion	Malignant Pleural Effusion	Drug: SCB-313	Phase 1	NCT04123886
		Completed	Study with SCB-313 (Recombinant Human TRAIL-Trimer Fusion Protein) for Treatment of Malignant Ascites	Malignant Ascites	Drug: SCB-313	Phase 1	NCT04051112
		Completed	Study with SCB-313 (Recombinant Human TRAIL-Trimer Fusion Protein) for Treatment of Malignant Pleural Effusions	Malignant Pleural Effusions	Drug: SCB-313	Phase 1	NCT03869697
		Completed	Study with SCB-313 (Recombinant Human TRAIL-Trimer Fusion Protein) for Treatment of Peritoneal Malignancies	Peritoneal Malignancies	Drug: SCB-313	Phase 1	NCT03443674
		Recruiting	A Phase I Study Evaluating SCB-313 for the Treatment of Subjects with Peritoneal Carcinomatosis	Peritoneal Carcinomatosis	Drug: SCB-313	Phase 1	NCT04047771
Human IgG (DR5 specific antibody)	Conatumumab	Completed	Study of Birinapant in Combination with Conatumumab in Subjects with Relapsed Ovarian Cancer	Relapsed Epithelial Ovarian Cancer	Drug: Birinapant	Phase 1	NCT01940172
				Relapsed Primary Peritoneal Cancer	Drug: Conatumumab		
				Relapsed Fallopian Tube Cancer			
	Conatumumab	Completed	Conatumumab/Panitumumab Combination Metastatic Colorectal Cancer Study	Colon Cancer	Drug: Panitumumab	Phase 1/2	NCT00630786
				Colorectal Cancer	Drug: Conatumumab		
				Rectal Cancer			
				Metastatic Colorectal Cancer			
				Oncology			
	Conatumumab	Completed	Open Label Extension Study of Conatumumab and Ganitumab (AMG 479)	Advanced Solid Tumors	Drug: Modified FOLFOX6	Phase 2	NCT01327612
				Carcinoid	Biological: Conatumumab		
				Colorectal Cancer	Biological: Ganitumab		
				Locally Advanced	Biological: Bevacizumab		
				Lymphoma			
				Metastatic Cancer			
				Non-Small Cell Lung Cancer			
				Sarcoma			
				Solid Tumors			
	Conatumumab	Withdrawn	Conatumumab, Gemcitabine Hydrochloride, Capecitabine, and Radiation Therapy in Treating Patients with Locally Advanced Pancreatic Cancer	Pancreatic Cancer	Biological: Conatumumab	Phase 1/2	NCT01017822
					Drug: Capecitabine		
					Drug: Gemcitabine hydrochloride		
					Radiation: 3-dimensional conformal radiation therapy		
	Conatumumab	Completed	QUILT-2.018: Safety & Efficacy of FOLFIRI with AMG 479 or AMG 655 vs. FOLFIRI Alone in KRAS-mutant Metastatic Colorectal Carcinoma	Metastatic Colorectal Cancer	Other: FOLFIRI	Phase 2	NCT00813605
					Biological: AMG 655		
					Other: Placebo		
					Biological: AMG 479		
	Conatumumab	Completed	Phase 1b/2 Study of AMG 655 with Doxorubicin for the First-Line Treatment of Unresectable Soft Tissue Sarcoma	Locally Advanced or Metastatic, Unresectable Soft Tissue Sarcoma	Drug: AMG 655	Phase 1/2	NCT00626704
				Sarcoma	Other: Placebo		
				Soft Tissue Sarcoma	Drug: Doxorubicin		
	Conatumumab	Terminated	QUILT-3.026: AMG 655 in Combination with AMG 479 in Advanced, Refractory Solid Tumors	Colorectal Cancer	Biological: AMG 479	Phase 1/2	NCT00819169
				Locally Advanced	Biological: AMG 655		
				Metastatic Cancer			
				Non-Small Cell Lung Cancer			
				Ovarian Cancer			
				Pancreatic Cancer			
				Sarcoma			
				Solid Tumors			
	Conatumumab	Completed	QUILT-2.019: A Study of AMG 655 or AMG 479 in Combination with Gemcitabine for Treatment of Metastatic Pancreatic Cancer	Adenocarcinoma of the Pancreas	Other: Placebo	Phase 1/2	NCT00630552
				Metastatic Pancreatic Cancer	Drug: AMG 479		
				Pancreatic Cancer	Drug: AMG 655		
	Conatumumab	Completed	A Phase 1b/2 Study of AMG 655 in Combination with Paclitaxel and Carboplatin for the First-Line Treatment of Advanced Non-Small Cell Lung Cancer	Non-Small Cell Lung Cancer	Drug: AMG 655	Phase 1/2	NCT00534027
					Other: AMG 655 placebo		
	Conatumumab	Completed	Phase 1b/2 Study of AMG 655 with mFOLFOX6 and Bevacizumab for First-Line Metastatic Colorectal Cancer	Metastatic Colorectal Cancer	Drug: Placebo	Phase 1/2	NCT00625651
				Colon Cancer	Drug: AMG 655		
				Colorectal Cancer	Drug: Modified FOLFOX6		
				Rectal Cancer	Drug: Bevacizumab		
	Conatumumab	Completed	Phase 1b Lymphoma Study of AMG 655 in Combination with Bortezomib or Vorinostat	Hodgkin’s Lymphoma	Drug: AMG 655	Phase 1	NCT00791011
				Low Grade Lymphoma	Other: Vorinostat		
				Lymphoma	Other: Bortezomib		
				Mantle Cell Lymphoma			
				Non-Hodgkin’s Lymphoma			
				Diffuse Large Cell Lymphoma			
Humanized IgG (DR5 specific antibody)	Tigatuzumab	Completed	Abraxane with or without Tigatuzumab in Patients with Metastatic, Triple Negative Breast Cancer	Breast Cancer	Drug: Abraxane alone	Phase 2	NCT01307891
				Triple Negative Breast Cancer	Drug: Abraxane + Tigatuzumab		
				Stage IV Breast Cancer			
				Metastatic Breast Cancer			
	Tigatuzumab	Completed	A Phase I Imaging and Pharmacodynamic Trial of CS-1008 in Patients with Metastatic Colorectal Cancer	Colorectal Neoplasms	Drug: CS-1008	Phase 1	NCT01220999
Humanized IgG (DR5 specific)	DS-8273a	Terminated	Antibody DS-8273a Administered in Combination with Nivolumab in Subjects with Advanced Colorectal Cancer	Colorectal Neoplasm	Drug: DS-8273a + nivolumab	Phase 1	NCT02991196
	DS-8273a	Completed	Open-label Study of DS-8273a to Assess Its Safety and Tolerability, and Assess Its Pharmacokinetic and Pharmacodynamic Properties in Subjects with Advanced Solid Tumors or Lymphomas	Advanced Solid Tumor	Drug: DS-8273a	Phase 1	NCT02076451
				Lymphoma			
	DS-8273a	Completed	Study of DS-8273a with Nivolumab in Unresectable Stage III or Stage IV Melanoma	Melanoma	Biological: DS-8273a	Phase 1	NCT02983006
					Biological: Nivolumab		
Human IgG (DR4 specific antibody)	Mapatumumab	Completed	Study of Mapatumumab in Combination with Sorafenib in Subjects with Advanced Hepatocellular Carcinoma	Carcinoma, Hepatocellular	Drug: Mapatumumab	Phase 1/2	NCT01258608
					Drug: Placebo		
					Drug: Sorafenib		
	Mapatumumab	Completed	Study of Mapatumumab in Combination with Bortezomib (Velcade) and Bortezomib Alone in Subjects with Relapsed or Refractory Multiple Myeloma	Multiple Myeloma	Biological: Mapatumumab	Phase 2	NCT00315757
					Drug: Bortezomib		
	Mapatumumab	Completed	Mapatumumab, Cisplatin and Radiotherapy for Advanced Cervical Cancer	Advanced Cervical Cancer	Drug: Mapatumumab	Phase 1/2	NCT01088347
					Drug: Cisplatin		
					Radiation: radiotherapy		
	Mapatumumab	Completed	A Study of Mapatumumab in Combination with Sorafenib in Subjects with Advanced Hepatocellular Carcinoma	Hepatocellular Carcinoma	Biological: Mapatumumab	Phase 1	NCT00712855
					Drug: Sorafenib		
	Mapatumumab	Completed	A Study of Mapatumumab in Combination with Paclitaxel and Carboplatin in Subjects with Non-small Cell Lung Cancer	Non-Small Cell Lung Cancer	Biological: Mapatumumab	Phase 2	NCT00583830
					Drug: Paclitaxel		
					Drug: Carboplatin		
	Mapatumumab	Completed	Study of TRM-1(TRAIL-R1 Monoclonal Antibody) in Subjects with Relapsed or Refractory Non-Small Cell Lung Cancer (NSCLC)	Carcinoma, Non-Small-Cell Lung	Drug: TRAIL-R1 mAb (TRM-1;HGS-ETR1)	Phase 2	NCT00092924
	Mapatumumab	Completed	Study of TRM-1 (TRAIL-R1 Monoclonal Antibody) in Subjects with Relapsed or Refractory Non-Hodgkin’s Lymphoma (NHL)	Lymphoma, Non-Hodgkin	Drug: TRAIL-R1 mAb (TRM-1; HGS-ETR1)	Phase 2	NCT00094848
Multivalent Nanobody (DR5 specific)	TAS266	Terminated	First in Human Trial of TAS266 in Patients with Advanced Solid Tumors	Advanced Solid Tumors	Drug: TAS266	Phase 1	NCT01529307
The bispecific, tetravalent fibroblast-activation protein (FAP)-TRAIL-R2 antibody	RO6874813 (RG7386)	Completed	A Dose Escalation Study of RO6874813 in Participants with Locally Advanced or Metastatic Solid Tumors	Neoplasms	Biological: RO6874813	Phase 1	NCT02558140
Hexabody (DR5 specific)	GEN1029	Terminated	GEN1029 (HexaBody^®^-DR5/DR5) Safety Trial in Patients with Malignant Solid Tumors	Colorectal Cancer	Biological: GEN1029 (HexaBody^®^-DR5/DR5)	Phase 1/2	NCT03576131
				Non-small Cell Lung Cancer			
				Triple Negative Breast Cancer			
				Renal Cell Carcinoma			
				Gastric Cancer			
				Pancreatic Cancer			
				Urothelial Cancer			
Fc-TRAIL	ABBV-621	Completed	A Study of the Safety and Tolerability of ABBV-621 in Participants with Previously-Treated Solid Tumors and Hematologic Malignancies	Advanced Solid Tumors	Drug: ABBV-621	Phase 1	NCT03082209
				Cancer	Drug: Venetoclax		
				Hematologic Malignancies	Drug: Bevacizumab		
					Drug: FOLFIRI		
	ABBV-621	Recruiting	Study to Determine Recommended Phase 2 Dose of Intravenous (IV) Eftozanermin Alfa in Combination with IV or Subcutaneous (SC) Bortezomib and Oral Dexamethasone Tablet and to Assess Change in Disease Symptoms in Adult Participants with Relapsed or Refractory Multiple Myeloma	Multiple Myeloma	Drug: Eftozanermin alfa (ABBV-621)	Phase 1	NCT04570631
					Drug: Bortezomib		
					Drug: Dexamethasone		
Multimeric Anti-DR5 IgM Agonist	IGM-8444	Recruiting	Phase I Study of IGM-8444 Alone and in Combination in Subjects with Relapsed, Refractory, or Newly Diagnosed Cancers	Solid Tumor	Drug: IGM-8444	Phase 1	NCT04553692
				Colorectal Cancer	Drug: FOLFIRI		
				Non-Hodgkin Lymphoma	Drug: Bevacizumab (and approved biosimilars)		
				Sarcoma	Drug: Birinapant		
				Chondrosarcoma	Drug: Venetoclax		
				Small Lymphocytic Lymphoma	Drug: Gemcitabine		
				Chronic Lymphocytic Leukemia	Drug: Docetaxel		
				Acute Myeloid Leukemia	Drug: Azacitidine		
Tetravalent DR5 agonistic antibody	INBRX-109	Recruiting	Study of INBRX-109 in Conventional Chondrosarcoma	Conventional Chondrosarcoma	Drug: INBRX-109	Phase 2	NCT04950075
					Drug: Placebo		
	INBRX-109	Recruiting	Phase 1 Study of INBRX-109 in Subjects with Locally Advanced or Metastatic Solid Tumors Including Sarcomas	Solid Tumors	Drug: INBRX-109	Phase 1	NCT03715933
				Malignant Pleural Mesothelioma	Drug: Carboplatin		
				Gastric Adenocarcinoma	Drug: Cisplatin		
				Colorectal Adenocarcinoma	Drug: Pemetrexed		
				Sarcoma	Drug: 5-fluorouracil		
				Pancreatic Adenocarcinoma	Drug: Irinotecan		
				Ewing Sarcoma	Drug: Temozolomide		
				Chondrosarcoma			
Mesenchymal stromal cells expressing TRAIL	MSC-TRAIL	Recruiting	Targeted Stem Cells Expressing TRAIL as a Therapy for Lung Cancer	Adenocarcinoma of Lung	Genetic: MSCTRAIL	Phase 1/2	NCT03298763
					Drug: Placebo		
Immunotherapy	CAR-T cell therapy	Recruiting	Autologous CAR-T/TCR-T Cell Immunotherapy for Solid Malignancies	Esophagus Cancer	Biological: CAR-T/TCR-T cells immunotherapy	Phase 1/2	NCT03941626
				Hepatoma			
				Glioma			

## Data Availability

Not applicable.

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
