# Peer review of "Targeting TRAIL Death Receptors in Triple-Negative Breast Cancers: Challenges and Strategies for Cancer Therapy"

_cells, 2022, doi:10.3390/cells11233717_

Round 1

Reviewer 1 Report

Overall, this is a well written and relevant review article. The manuscript is mechanistically focused, which is good. However, the title of the manuscript and its abstract give the impression that the review will focus on Triple Negative Breast Cancer, but after reading it through, that is not the impression a reader gets. It is very important to include other cancers in this review as it might explain mechanisms, treatment results, etc. However, for every section, there should be a separate subsection focusing solely on TNBC. Since there are other reviews focusing on TRAIL receptor-targeted therapy (such as this one https://doi.org/10.2217/14796694.2.4.493), it is important to differentiate this review and have it focused on TNBC.

In addition, information can be a little confusing. In section 2, it is mentioned that TRAIL activation results in tumor suppression and that TRAIL depletion results in enhanced tumor growth and metastasis. However, in the last paragraph of section 3, it is said that TRAIL expression has been shown to positively regulate cell growth. So, it looks like TRAIL can have both tumor suppression and activation effects. Perhaps it will be good to mention this first and then organize which conditions might contribute to tumor repression or tumor promotion.

It was also unclear to me if the authors wanted to mainly focus on clinical results. The introduction seems to contain mostly preclinical data and information regarding TRAIL and DR pathways. However, the other sections seem to focus mostly on clinical data. I believe that his paper would be better organized it the authors divided each section into preclinical vs clinical data. That will facilitate the reading: preclinical data found this, but clinical data did not reveal that, etc.

I also believe that the use of TRAIL agonists with other therapies (combinational therapies) should be expanded to provide further details in mechanisms, why it is potentially synergistic, what has been found in preclinical data and what was hold true in clinical data. Combination with radiotherapy, for example, should also be included, as well as other therapies such as immunotherapy.

In addition, I believe the authors can move their supplementary tables to the main manuscript. There are only two figures in this review. The authors can also consider adding more figures with data (can be preclinical results) on anti-tumor results and mechanisms of action.

Reviewer 2 Report

This work addresses the possibility of exploiting the TNF-related apoptosis inducing ligand (TRAIL) pathway to specifically trigger apoptosis in cancer cells. The authors provide a very clear overview of how TRAIL works at the molecular level and a very comprehensive state-of-the art of its (pre)clinical potential, both as in vitro/in vivo work as well as in terms of recent or ongoing clinical trials.  In particular, attention was placed on the possible use of  Death Receptor (DR) agonists as a means of enhancing cancer cell apoptosis. The conceptual framework is well designed: after examining clinical trial results, pharmacokinetics and pharmacodynamics of used agonists were reviewed to possibly explain the lack of efficacy in parallel to predictive biomarkers to use in order to target those patients that promise a better response. The authors then move on to discuss in detail several strategies to improve delivery and targeting of the TRAIL/DR pathway by the aforementioned DR agonists. 

One criticism, however, is that  the reader may not well understand why the focus of such an effort was stated to be on triple negative breast cancer (TNBC): apart from the abstract and the Introduction, were the peculiarly poor prognosis of TNBC patients is highlighted, in all other Sections one fails to see such a focus. Indeed, examples of  results of trials, preclinical studies and molecular mechanisms specifically applied to TNBC are cited, but amid a wealth of information encompassing a variety of other solid cancers. Therefore, considering the effort the authors have produced in compiling such a rich review, I would strongly suggest to re-phrase/remove the focus on TNBC to make their work valuable in a more general sense. This is, indeed,  the underlying message of the final paragraph (Conclusions and Future Directions), where no defined confinement of their work is mae to breast cancer types.

Two very minor mistakes: 

Line 237: The sentence "...results from clinical trials have been largely failed..." should read "...results from clinical trials have largely failed..."

Lines 296-297: similarly, the sentence "...recent work has evaluating positron emission tomography imaging-based sensors" should read "..recent work has evaluated positron emission tomography imaging-based sensors..."

Reviewer 3 Report

This is a very interesting, serious and fundamental analytical review. The review is undoubtedly worthy of publication in the desired journal. I have only one wish. Perhaps the authors would consider it necessary to prepare a separate figure reflecting the available methods for delivering TRAIL activators to cells.

Round 2

Reviewer 1 Report

The authors addressed my comments and the manuscript is significantly improved.